# Healthcare providers' perceptions of the kidney transplant program at King Faisal Hospital Rwanda: A one-year evaluation

Marissa Martinelli[1]*, Emile Twagirumukiza[2], Ahmed M. Elbasha[2],
Augustin Sendegeya[2,3], Belise S. Uwurukundo[2,3], Jules Karangwa[2], Kara L. Neil[2,3],
Laetitia Nshimiyimana[2], Jeffrey Punch[4]

1 University of Michigan Medical School, University of Michigan, Ann Arbor, Michigan, United States of America, 2 King Faisal Hospital Rwanda, Kigali, Rwanda, 3 Africa Health Sciences University, Kigali, Rwanda, 4 Transplant Surgery, University of Michigan, Ann Arbor, Michigan, United States of America

* marismar@umich.edu

## Abstract

The living donor kidney transplant program was established at King Faisal Hospital Rwanda (KFH) in 2023, to reduce medical abroad referrals and strengthen the health care delivery system. The study aimed to explore healthcare provider perspectives, their needs for further developing the kidney transplant program at KFH and provide recommendations to fill these gaps. This study employed a sequential explanatory mixed methods approach embedded within an overarching exploratory research design, collecting data from healthcare workers who participated in at least one kidney transplant mission at KFH since 2023. Participants represented the dialysis, outpatient renal clinics, operating theatres, and postoperative kidney transplant units. Data were collected through initial surveys, which informed the semi-structured interviews. Descriptive and thematic analyses of the results were performed. The study included 50 respondents. 32% of respondents were extremely satisfied with the devices, infrastructure, and technology used in kidney transplantation at KFH, while also positing that missing or non-functional devices limit their ability to care for kidney transplant patients. 11 healthcare providers were interviewed. The most reported strength of the program was effective governance and management systems, while the areas needing improvement were education and training, patient care coordination, infrastructure, instruments, and devices. The findings of this study underscore the importance of strong political and institutional will, hospital organization, and multidisciplinary team collaboration in sustaining renal transplantation programs. Participants recommended practical-based training, rotations into high-volume facilities, and regular refresher training. Early renal disease diagnosis and ensuring treatment affordability were also recommended.

**Data availability statement:** Study data, including survey results, are all included in the paper and its Supporting Information files.

**Funding:** The study was funded by the University of Michigan Medical School Capstone for Impact (CFI) Grant (https://medschool.umich.edu/programs-admissions/md-program/md-curriculum/impact-curriculum) and the University of Michigan Global REACH CFI Supplemental Grant (https://medschool.umich.edu/centers-institutes/global-reach/what-we-do/student-funding). Both awards were received by MM. The funders had no role in study design, data collection and analysis, decision to publish, or preparation of the manuscript.

**Competing interests:** The authors have declared that no competing interests exist.

## Introduction

While health outcomes in Rwanda have been greatly improving over the last two decades, noncommunicable diseases, including chronic kidney disease, were estimated to account for 44% of all mortality in the country in 2016 [1]. Diabetes and hypertension are significant risk factors for chronic kidney disease. In 2022, their prevalence was 2.9% nationwide (9.8% in Kigali city), and 16.8%, respectively. As of 2024, Rwanda had approximately 250 chronic renal dialysis patients, with 64 machines available [2]. Historically, patients have been referred abroad for kidney transplant surgery, resulting in a significant financial burden for both the patients and the Government of Rwanda [2]. Kidney transplantation has globally been shown to reduce mortality and improve quality of life compared to dialysis for chronic kidney disease patients [3].

The clear indication for a sustainable kidney transplant program in Rwanda has motivated ongoing collaborative efforts to train healthcare workers at King Faisal Hospital Rwanda (KFH) in all aspects of transplant care. The first kidney transplant was completed in May 2023, and a total of 32 living donor transplants have been completed as of July 2024. The development of any new program at a healthcare institution, especially with the complexity involved in transplant care, comes with anticipated and unanticipated challenges [4].

This assessment of the kidney transplant (KT) program aimed to identify progress made to date and define existing needs by examining healthcare providers' perspectives, which provides a basis for future collaborative solutions to fill these gaps.

## Materials and methods

### Ethics statement

The study was approved by King Faisal Hospital Rwanda IRB (REF: KFH/2024/217/IRB). A written informed consent form was obtained from all interview participants. Participation in the study was voluntary.

### Setting and design

The study utilized a sequential, explanatory mixed methods approach. Data was collected from an initial survey and semi-structured interviews. The first phase of the initial survey was made of the set of questions to quantitatively assess the role, duration, and brief scaling of the experience with the kidney transplant program, identify areas where major challenges are faced, and confirm the willingness to proceed with in-depth experience exploration. The second phase of interviews was informed by the responses from the initial phase. The interview guide was customized to exclusively and exhaustively gather perceptions of the healthcare professionals across all roles. The study setting KFH, is a multispecialty quaternary hospital located in Kigali, Rwanda. It provides dialysis and living donor kidney transplantation to Rwandan and neighboring populations with end-stage renal disease (ESRD). Planning for the renal transplant surgery program and fellowship programs in nephrology and transplant surgery through the University of Rwanda began with virtual meetings in 2021 that

were held monthly with participants from existing U.S. kidney transplant providers. A site visit occurred in late 2022, and the first kidney transplants occurred in May 2023 [2,5].

### Study participants

Study participants included healthcare professionals, support staff, and administrators involved in the kidney transplant program at KFH since May 2023. Inclusion criteria were participation in at least one mission since the start of the transplant program. Exclusion criteria were lack of participation in the transplant program and involvement as a patient at KFH. Participation in the study was voluntary. The recruitment period was June 11, 2024 through June 26, 2024.

### Study instruments and data collection

A questionnaire was distributed via email and WhatsApp to the eligible participants via Google Forms. In addition, participation was solicited in-person at various departments, with regular reminders sent. The variables of the survey tool included the general characteristics of the participants, their perceptions toward infrastructure, devices, and medical equipment utilized in kidney transplantation and areas of improvement.

Semi-structured interview participants were also sampled from survey respondents via random stratified sampling according to the role and units. Interviews included open-ended questions, as well as specific questions informed by survey responses and observations to further explore their perceptions and experiences. Written informed consent was obtained from each participant prior to conducting these audio-taped interviews. This study was approved by the KFH Institutional Review Board (Ref. #KFH/2024/217/IRB).

### Inclusivity in global research

Additional information regarding the ethical, cultural, and scientific considerations specific to inclusivity in global research is included in the Supporting Information (S1 Checklist).

### Data analysis

Data from the questionnaire were analyzed quantitatively. Descriptive analyses were performed, and results were reported in tables and graphs as frequencies and percentages. The recorded audio interviews were transcribed into text. Two researchers conducted transcript cleaning, reviewing, and cross-checking to ensure the completeness of the data. Dedoose software was used to inductively generate codes, pull quotations, and categorize themes for the thematic analysis approach. Codes were later merged into themes and subthemes.

## Results

Out of 65 participants who were involved in the kidney transplant program, 50 completed the online questionnaire, representing a range of roles within the program. Most of the respondents were nurses (n = 20) followed by the non-surgical physicians (n = 12) (Table 1).

As shown below, most of the respondents somewhat (52%) or strongly (32%) agreed that they were satisfied with the transplant program devices, infrastructure, and technology (Table 2 and S1 Table).

A lack of staff training was the most common challenge indicated by 50% of the respondents. Other areas in which many participants reported difficulties included the pre-operative laboratory setting, reviewing radiology images, and intra-operative instrumentation (Table 3).

A total of 11 semi-structured interviews were conducted. Two interviews were conducted as a group with 2 participants each, and the rest were individual interviews. The names of the interviewees were blinded, and unique identifiers were created (Table 4).

During the analysis of the interview records, three main themes emerged, including strong governance and management systems, education and training, and patient care coordination.

**Table 1. Questionnaire participation by role.**

| Role | Number |
|---|---|
| Nurses | 21 |
| Physicians (Non-surgeons) | 12 |
| Surgeons | 5 |
| Facilities operators | 5 |
| Non-Physician Anesthetists | 3 |
| Administrators | 2 |
| Allied health professionals | 2 |

**Table 2. Number of participants in agreement with the statements on the survey.**

| Statement | Strongly Disagree n (%) | Somewhat Disagree n (%) | Somewhat Agree n (%) | Strongly Agree n (%) | Not Applicable n (%) |
|---|---|---|---|---|---|
| I am satisfied with the infrastructure needed for the kidney transplant program. | 0 (0) | 4 (8) | 26 (52) | 16 (32) | 4 (8) |
| I am satisfied with the medical devices used in the kidney transplant program. | 0 (0) | 2 (4) | 14 (28) | 26 (52) | 7 (14) |
| I am satisfied with the technology used in the kidney transplant program. | 1 (2) | 5 (10) | 19 (38) | 20 (40) | 5 (10) |
| There are medical devices necessary for my role that are difficult or confusing to use. | 15 (30) | 8 (16) | 13 (26) | 5 (10) | 9 (18) |
| There are software or computer applications necessary for my role that are difficult/confusing to use. | 17 (34) | 7 (14) | 7 (14) | 7 (14) | 12 (24) |
| There are times that the devices I am using do not function the way I need them to. | 16 (32) | 10 (20) | 10 (20) | 9 (18) | 5 (10) |
| There are times I do not know how to use a device in my job. | 20 (40) | 5 (10) | 12 (24) | 6 (12) | 7 (14) |
| There are times that I do not have access to equipment necessary for my role. | 20 (40) | 7 (14) | 9 (18) | 6 (12) | 8 (16) |
| Missing or non-functional devices limit my ability to care for kidney transplant patients. | 15 (30) | 6 (12) | 11 (22) | 7 (14) | 11 (22) |
| Issues with hospital infrastructure delay or limit the care of kidney transplant patients. | 15 (30) | 10 (20) | 10 (20) | 10 (20) | 5 (10) |

## Theme 1: Strong governance and management systems

Respondents identified the strengths of the kidney transplant program, which are more related to the supportive national and hospital leadership, a conducive working environment, and a highly committed transplant team. Several respondents highlighted the ongoing government support in optimizing the outcomes of the program. The hospital leadership is well-organized and committed to ensuring program effectiveness and sustainability.

> *"It's a program that is being done here, and it's being fully supported by the government. So that's why we have, I think many doors for us are very open. They supported the training of surgeons and nephrologists; they are supporting the logistics, and the patients either for the transplant and the follow-up. They are also planning for sustainability so that after few years, we could be able to run on daily basis, not just on a mission basis." (Nephrologist)*

**Table 3. Areas in which survey participants experience challenges in infrastructure, devices, or technology.**

| Area | Participants n (%) |
|---|---|
| Staff training | 25 (50) |
| Other | 13 (26) |
| Pre-operative laboratory testing | 12 (24) |
| Reviewing radiology images | 10 (20) |
| Intra-operative instrumentation | 10 (20) |
| Infection control | 7 (14) |
| Intraoperative patient positioning | 6 (12) |
| Organ perfusion | 5 (10) |
| Training on surgical techniques | 5 (10) |
| Post-operative fluid management | 3 (6) |
| Transplant surgery induction | 1 (2) |

**Table 4. Interview participation per department.**

| Department | Number |
|---|---|
| Biomedical engineering | 2 |
| Nephrology | 1 |
| Operating theatre | 6 |
| Dialysis | 1 |
| Intensive care unit | 1 |

Respondents reported successful teamwork in which everyone is responsible and committed to providing high-quality care for patients.

*"We work as a team to have a well-prepared patient. And even through the whole procedure, we work as a team, we collaborate. The kidney transplant, it's a very good program which is going to help Rwandans, not only Rwandans, but also neighboring countries." (Non-physician anesthetist)*

*"Teamwork is another thing that I can say that is just improving and it is a good thing for the whole team. And also, we have fellows in different specialties like surgery and nephrology that are going to start helping us a lot in the future." (Transplant Nurse)*

### Theme 2: Strengthening education and training

Most of the interviewees reported that transplant-related skills development and staff experience are the most common issues at KFH. Operating theater nurses indicated not having formal training in kidney transplant surgeries. The surgeons have a fellowship program, but they indicated that less emphasis was given to formalized nurse training.

*"We don't have training. So what we are doing one way we are basing on what Professor (name blinded) taught us, but also it's us struggling to go for Google, for YouTube. So we don't have training on it." (Theatre Nurse)*

The respondents argued that training transplant surgeons takes a long time because the fellows are only able to do two transplants every couple of months, so the need for regular training and practice of both surgeons and nurses was emphasized.

*"We did the procedure [for] three months and the other three months without these procedures. After three months, we start on zero. So, it's better to work closer… time to go where you do the transplant day to day and…[are] familiar [with] the procedures." (Theatre Nurse)*

**Theme 3: Patient care coordination**

Timely and coordinated care for ESRD patients effectively delays the disease progression, thus improving treatment outcomes and reducing healthcare costs [6]. A way to identify chronic kidney disease in patients in Rwanda at an earlier stage was identified by participants as an opportunity to treat patients before the disease progresses.

*"We are still having [issues] identifying chronic kidney disease at [an] early stage and following them. So, because we have many patients who come who are [at the] end-stage already and need to start dialysis." (Nephrologist)*

Respondents indicated a concern about the disproportion between the capacity of the programs versus the number of patients in need of renal transplants.

*"...according to the population we have with end-stage renal disease, I think we should have a long list. But now we [have the] capacity to operate [to] one patient at a time." (Theatre nurse)*

Globally, many chronic kidney disease patients cannot afford dialysis, which participants highlighted as a limitation for access to care. Community-based health insurance (CBHI), which covers more than 80% of the Rwandan population [6], provides financial support for acute kidney injury for 6 weeks, but not for chronic kidney disease. They also cover patients who have been authorized for transplant, but not during the evaluation process. Similarly, transplanted patients need life-long immunosuppressant medications.

*"Patients who come here, some less than 20%, maybe not doing dialysis three times a week, but they do like two times a week. I [can] say that some of our patients do not do dialysis, like for the whole week because of financial problems." (Dialysis Nurse)*

## Discussion

This study employed a mixed methods approach to evaluate healthcare provider perceptions regarding the needs of the kidney transplant program at KFH, since its establishment. Several strengths of the program were identified, including effective teamwork, the location of the program in Rwanda, where patients are taken care of by their people, effective leadership and support from the administration, and recognition of the value of the program by all team members.

The respondents in this study indicated that they were motivated by seeing the positive impact of the KT program on the patient's quality of life. This aligns with various reports that the goals of kidney transplants are to optimize the quality of life and increase the life expectancy of ESRD patients [7,8].

This study demonstrated that reliable technical and financial support at the national and institutional levels enables the kidney transplant program to run smoothly and promises its sustainability. A survey by the International Society of Nephrology (ISN) indicated that national oversight systems for renal health services are more prevalent in high-income

countries, while lower-income countries have hospital oversight. In lower resource settings, health financing typically competes with other priorities, and a linear increase of the health budget with the gross domestic product per capita has been reported [9]. Kidney transplant services are publicly funded in 57% of global countries, with a much higher private and out-of-pocket rate in African countries [10]. In the case of Rwanda, participants indicated that both of these systems are in place, making it an ideal place for kidney transplant surgery.

Unmet needs were broadly identified in areas of patient care coordination, and education and training. A study reported that some ESRD patients had barriers to dialysis compliance due to individual financial barriers and restrictive insurance policies. These findings align with a 2018 study conducted in Rwanda, which indicated that 34% of the patients were receiving two dialysis sessions per week [11]. Currently, thrice-weekly dialysis remains the standard of care for survival and quality of life, according to clinical practice guidelines [12,13]. Even though some insurance schemes cover up to 100% of the dialysis services, the study underlines the need for health insurance providers to continually review dialysis accessibility.

Many patients at KFH utilized catheters for hemodialysis access because they relied on visiting vascular surgeons for fistula creation procedures. Patients with catheters have higher rates of catheter-associated bloodstream infections, leading to decreased quality of life and more time spent in hospitals. Consistent results were reported in many studies conducted in African countries [4,14,15], and underline the importance of integrating arteriovenous (AV) fistula creation procedures into the transplant fellowship training during transplant weeks, with the additional benefit of vascular surgical technique practice for fellows.

Study respondents emphasized a growing need for education and training of service providers contributing to the program. Some staff had been exposed to high-volume renal transplant hospitals in India, South Africa, and the US, but others had never had this opportunity. With the transplantation surgery conducted every couple of months, both transplant surgery fellows, anesthetists, and nurses required refresher continuous development programs and simulation-based training. According to Stefanidis et al., technical skills acquired from simulation-based training are often transferred to the operating room [16].

There is currently no locally offered formal training in transplant surgery for KFH operating theatre nurses. The nurses involved in the program gained skills through clinical practice and mentorship. Participants indicated that there is a need for a way to improve formal, certified training in kidney transplant surgeries for operating theatre nurses. One recommendation is for participating nurses to briefly review the procedures together on the Monday of each transplant week. A printed training guide can be developed with steps of both the donor and recipient surgeries, including possible complications to anticipate. The guide should include pictures of the instruments and set-up for the procedures, importantly including uncommon instruments that are only used in complications.

Having a higher volume of access to transplants was highlighted as a recommendation for fellows to maintain surgical technical skills. In addition, transplant fellows are urologists due to their expertise in nephrectomies and ureter procedures, but have limited vascular surgery experience that is crucial for the anastomoses in renal transplants. Therefore, it would be important for KFH transplant fellows to practice renal transplant surgery techniques in between transplant weeks to improve the efficiency of their training and develop confidence in performing these procedures independently in the future. The University of Michigan Transplant Surgery team produces 3D-printed models of a kidney transplant field for their residents to practice technical skills of arterial and venous anastomoses [17]. This model could be expanded upon to include online modules and assessments for an integrated self-training system specific to kidney transplants.

## Strengths and limitations of the study

There were some limitations to this study. Specifically, the findings are specific to the institution and may not be generalizable. Additionally, there is an imbalance of professions represented in our participant pool. Input from roles with fewer representatives (for example, administrative staff accounted for 2/50 questionnaire participants and 0/11 interviewees)

could be lost among the clinicians' voices. Additionally, there may also be biases, as the healthcare providers are the ones reporting on the program they implement themselves.

## Conclusion

Significant progress has been made in the kidney transplant program at KFH within only one year. This is an important opportunity for the country to improve care for patients with chronic kidney disease. A needs assessment conducted to evaluate the current status of the program revealed strengths, including strong teamwork, effective leadership, and government support. Several unmet needs were identified within the areas of patient care coordination, education and training. Knowledgeable and skilled team members are the most valuable resource of the program. Emphasis may be on training opportunities for all team members, especially in the current setting of infrequent transplant surgeries. These insights and recommendations may be valuable for the development of kidney transplant programs at similar institutions.

## Supporting information

**S1 Checklist. Inclusivity in global research.**
(PDF)

**S1 Table. KFH kidney transplant program initial survey - organized responses.**
(XLSX)

## Acknowledgments

The authors would like to thank the management and staff of the King Faisal Hospital Rwanda, and the University of Michigan for the invaluable support provided.

## Author contributions

**Conceptualization:** Marissa Martinelli.

**Data curation:** Marissa Martinelli, Emile Twagirumukiza.

**Formal analysis:** Emile Twagirumukiza, Kara L. Neil.

**Funding acquisition:** Marissa Martinelli.

**Investigation:** Marissa Martinelli, Emile Twagirumukiza, Belise S. Uwurukundo, Kara L. Neil, Laetitia Nshimiyimana, Jeffrey Punch.

**Methodology:** Marissa Martinelli, Emile Twagirumukiza, Ahmed M. Elbasha, Augustin Sendegeya, Belise S. Uwurukundo, Jules Karangwa, Kara L. Neil, Laetitia Nshimiyimana, Jeffrey Punch.

**Project administration:** Marissa Martinelli, Emile Twagirumukiza.

**Supervision:** Jeffrey Punch.

**Writing – original draft:** Marissa Martinelli, Emile Twagirumukiza, Kara L. Neil.

**Writing – review & editing:** Marissa Martinelli, Emile Twagirumukiza, Ahmed M. Elbasha, Augustin Sendegeya, Belise S. Uwurukundo, Jules Karangwa, Kara L. Neil, Laetitia Nshimiyimana, Jeffrey Punch.

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
