## [Decision Letter · Decision Letter 0]

PGPH-D-25-00853

Healthcare providers’ perceptions of the kidney transplant program at King Faisal Hospital Rwanda: A one-year evaluation

Dear Dr. Martinelli,

Thank you for submitting your manuscript to PLOS Global Public Health. After careful consideration, we feel that it has merit but does not fully meet PLOS Global Public Health’s publication criteria as it currently stands. Therefore, we invite you to submit a revised version of the manuscript that addresses the points raised during the review process.

Thank you for your very informative qualitative study on healthcare providers’ perceptions of the kidney transplant program at King Faisal Hospital Rwanda. Kindly apply all the revisions required by the reviewers so that this can proceed to publication. These are mostly minor, but are all essential to move this forward.

Consider the study sequence clarifications requested and addressing redundacy/repetition in discussion alongside other issues identified.

We look forward to receiving your revised manuscript.

Kind regards,

Barnabas Tobi Alayande

Academic Editor

Journal Requirements:

1. We have noticed that you have uploaded Supporting Information files, but you have not included a list of legends. Please add a full list of legends for your Supporting Information files after the references list.

Additional Editor Comments (if provided):

Reviewers' comments:

Reviewer's Responses to Questions

**Comments to the Author**

1. Does this manuscript meet PLOS Global Public Health’s publication criteria ? Is the manuscript technically sound, and do the data support the conclusions? The manuscript must describe methodologically and ethically rigorous research with conclusions that are appropriately drawn based on the data presented.

Reviewer #1: Yes

Reviewer #2: Yes

2. Has the statistical analysis been performed appropriately and rigorously?

Reviewer #1: Yes

Reviewer #2: Yes

3. Have the authors made all data underlying the findings in their manuscript fully available (please refer to the Data Availability Statement at the start of the manuscript PDF file)?

Reviewer #1: Yes

Reviewer #2: No

4. Is the manuscript presented in an intelligible fashion and written in standard English?

Reviewer #1: Yes

Reviewer #2: Yes

5. Review Comments to the Author

Reviewer #1: The manuscript is well written and results and well presented and discussed. However, some minor modifications are need in the methods section where the author reported that they used an exploratory research design. When explained the data collection procedures, it is unclear if they started with interviews and the results influenced the development of a quantitative tool as it should be in an exploratory mixed method design. Additionally, they way results are presented do not reflect an exploratory design rather, they show that data were collected concurrently and results were triangulated to meet the set objectives. Neither the qualitative method nor the quantitative method was developed following the results of the other. But, considering what's written on line 110 you may think that this was an explanatory mixed method design rather than an exploratory mixed method design.

Additionally, the exclusion criteria need to be explained and ethical considerations on participation whether it was voluntary or not. Also, some abbreviations need to be written in full before being used in the next sentences.

Reviewer #2: Overall Impression:

This manuscript addresses an important and timely topic: evaluating the progress and needs of a kidney transplant program from the perspective of healthcare providers and administrative staff.. The work is relevant and offers valuable insights, but some aspects require clarification and refinement to strengthen the paper’s rigor and transparency.

Major Comments:

• Abstract: Please correct the redundancy in the sentence "needs for further developing the developing kidney transplant program." of the background section.

• Discussion: Consider reorganizing parts of the discussion to group similar points together and reduce some repetition, particularly regarding challenges with dialysis services.

• Participant Composition: The authors mention that 20 of 50 respondents were nurses. However, a full breakdown of the professions (e.g., physicians, allied health professionals, administrative staff) is missing. Providing a complete distribution would allow readers to better assess the representativeness of the sample and understand potential professional biases in the results. A simple descriptive text would address this issue.

Minor Comments:

• Small language and grammar corrections are needed throughout

• I would suggest the authors to address the professional distribution imbalance in the Limitations section. For example, administrative staff are fewer (2/50), their responses could be lost among the clinicians’ voices.

6. PLOS authors have the option to publish the peer review history of their article (what does this mean? ). If published, this will include your full peer review and any attached files.

**Do you want your identity to be public for this peer review?** For information about this choice, including consent withdrawal, please see our Privacy Policy .

Reviewer #1: No

Reviewer #2: No

---

## [Decision Letter · Decision Letter 1]

PGPH-D-25-00853R1

Healthcare providers’ perceptions of the kidney transplant program at King Faisal Hospital Rwanda: A one-year evaluation

Dear Dr. Martinelli,

Thank you for submitting your manuscript to PLOS Global Public Health. After careful consideration, we feel that it has merit but does not fully meet PLOS Global Public Health’s publication criteria as it currently stands. Therefore, we invite you to submit a revised version of the manuscript that addresses the points raised during the review process.

Thank you for your clear revisions.

One reviewer has pointed out an important concept that needs to be clarified through the abstract and the body of the manuscript. Authors should very clearly distinguish the exploratory research design (in contrast to a conclusive design) from the explanatory mixed methods design that was used. The mixed methods is explanatory, not exploratory, and that is confounded in some aspects of the manuscript.

Abstract- "Methods: This study employed an exploratory design, with a mixed-methods approach" this can be confusing (easily misunderstood to speak to the mixed methods) and can be clarified- "This study employed a sequential explanatory mixed methods approach embedded within an overarching exploratory research design." (if you want to keep the concept of an exploratory study)

Line 45

Line 85-87 (also confusing) "The study was exploratory research by design. Data was collected from an initial survey and 87 semi-structured interviews" the sequence suggests that this is the definition of exploratory. Please clarify that this was a sequential explanatory mixed methods approach.

This is the only issue that is pending. Thank you for your prompt attention to this matter for clarity.

Kindly respond to the reviewer comments and the above to permit progress.

We look forward to receiving your revised manuscript.

Kind regards,

Barnabas Tobi Alayande

Academic Editor

Journal Requirements:

Additional Editor Comments (if provided):

Reviewers' comments:

Reviewer's Responses to Questions

**Comments to the Author**

1. If the authors have adequately addressed your comments raised in a previous round of review and you feel that this manuscript is now acceptable for publication, you may indicate that here to bypass the “Comments to the Author” section, enter your conflict of interest statement in the “Confidential to Editor” section, and submit your "Accept" recommendation.

Reviewer #1: (No Response)

Reviewer #2: All comments have been addressed

2. Does this manuscript meet PLOS Global Public Health’s publication criteria ? Is the manuscript technically sound, and do the data support the conclusions? The manuscript must describe methodologically and ethically rigorous research with conclusions that are appropriately drawn based on the data presented.

Reviewer #1: Yes

Reviewer #2: Yes

3. Has the statistical analysis been performed appropriately and rigorously?

Reviewer #1: Yes

Reviewer #2: Yes

4. Have the authors made all data underlying the findings in their manuscript fully available (please refer to the Data Availability Statement at the start of the manuscript PDF file)?

Reviewer #1: Yes

Reviewer #2: Yes

5. Is the manuscript presented in an intelligible fashion and written in standard English?

Reviewer #1: Yes

Reviewer #2: Yes

6. Review Comments to the Author

Reviewer #1: The author has addressed previous comments and rewrote the manuscript in a more understandable fashion to the audience, and the study identified the gaps in previous literature and how this study is contributing to show how the program is impacting the community and hospital staff. However, there is a comment that was not addressed properly. The author reported that they did an exploratory design but the description of the methodology in this study is not an exploratory design. I believe if this is not addressed properly it may confuse the audience since the description given matches with an explanatory sequential mixed method design. Basically, an exploratory mixed method design will start with a qualitative study and the results inform the quantitative study to explore or quantify the problem that was described in the qualitative study. I may suggest the author to clarify on the used terminology.

Reviewer #2: All comments have been addressed.

7. PLOS authors have the option to publish the peer review history of their article (what does this mean? ). If published, this will include your full peer review and any attached files.

**Do you want your identity to be public for this peer review?** For information about this choice, including consent withdrawal, please see our Privacy Policy .

Reviewer #1: No

Reviewer #2: No

---

## [Editor Report · Decision Letter 2]

Healthcare providers’ perceptions of the kidney transplant program at King Faisal Hospital Rwanda: A one-year evaluation

PGPH-D-25-00853R2

Dear Ms. Martinelli,

Thank you for the very direct and clear changes in response to the reviewer's concerns.

We are pleased to inform you that your manuscript 'Healthcare providers’ perceptions of the kidney transplant program at King Faisal Hospital Rwanda: A one-year evaluation' has been provisionally accepted for publication in PLOS Global Public Health.

Best regards,

Barnabas Tobi Alayande

Academic Editor